# Globalizing manifold-based reduced models for equations and data

**Bálint Kaszás** ✉ **& George Haller**

One of the very few mathematically rigorous nonlinear model reduction methods is the restriction of a dynamical system to a low-dimensional, sufficiently smooth, attracting invariant manifold. Such manifolds are usually found using local polynomial approximations and, hence, are limited by the unknown domains of convergence of their Taylor expansions. To address this limitation, we extend local expansions for invariant manifolds via Padé approximants, which re-express the Taylor expansions as rational functions for broader utility. This approach significantly expands the range of applicability of manifold-reduced models, enabling reduced modeling of global phenomena, such as large-scale oscillations and chaotic attractors of finite element models. We illustrate the power of globalized manifold-based model reduction on several equation-driven and data-driven examples from solid mechanics and fluid mechanics.

Reduced-order modeling is a widespread technique that seeks to simplify high-dimensional nonlinear systems while retaining their essential dynamical features. Among reduced-order modeling procedures, manifold-based methods have been steadily gaining momentum. This can be largely attributed to the prevalence of data-driven approaches that successfully build on the fundamental manifold hypothesis[1–3] of machine learning.

Center manifold reduction[4], geometric singular perturbation theory[5,6] and inertial manifold theory[7,8] all rely on the existence of low-dimensional attracting invariant manifolds in the phase space of a dynamical system. These methods constitute mathematically rigorous examples of nonlinear model reduction and yield truly predictive models. However, for systems encountered in practice that are not close to bifurcations, invariant manifolds can only be realistically constructed when they emanate from a known robust stationary state, such as a hyperbolic fixed point.

In those cases, seeking the invariant manifolds perturbatively and expressing them as local Taylor expansions at the known stationary state is justified. Traditionally, only stable and unstable manifolds, as continuations of the stable and unstable subspaces of the linearization, were approximated in this fashion[4]. The recent theory of spectral submanifolds (SSMs) extends this approach to arbitrary non-resonant spectral subspaces of the linearized system[9]. In particular, SSMs are now known to exist as smooth continuations of stable subspaces (like-

mode SSMs) and of subspaces spanned by stable and unstable modes (mixed-mode SSMs)[10]. This model reduction approach has been used in a broad range of physical settings to deduce very low-dimensional, mathematically justified polynomial models[9,10].

SSM reduction has been successfully applied to obtain accurate reduced models of nonlinear vibrations observed in high-dimensional finite element models[11,12] and experiments[13,14], multistable fluid flows[15,16], chaotic systems[17], fluid-structure interaction problems[18,19] and control of soft robots[20]. In an equation-driven setting, SSM reduction starts with the solution of an invariance equation through local Taylor expansions[11,21,22]. Data-driven SSM reduction[13] also uses polynomial expansions to obtain an approximation for SSMs.

SSMs are ideal tools for model reduction because their existence, uniqueness, and smoothness properties are precisely known. Specifically, if the governing equations are analytic, SSMs of attracting fixed points are guaranteed to also possess convergent Taylor series near the fixed point. However, the domain of convergence is generally unknown.

This has been a general limitation of invariant manifold-based reduction methods, restricting the range of applicability of the resulting reduced-order model to an a priori unknown domain. Importantly, the convergence-limiting singularity of the Taylor expansion is not a physical singularity of the dynamical system, and hence, invariant manifolds continue to exist even outside the domain

Institute for Mechanical Systems, ETH Zürich, Zurich, Switzerland. ✉e-mail: bkaszas@ethz.ch

of convergence of the Taylor expansions for those manifolds. Therefore, the breakdown of convergence arises without any prior indication. This major limitation of local approximation methods of invariant manifolds restricts the user to potentially small physical domains.

Here, we overcome this limitation by extending the local information contained in the Taylor series to considerably larger domains via the process generally known as analytic continuation[23]. Among commonly used methods of analytic continuation, we focus on Padé approximants[24]. Padé approximants are rational functions whose Taylor expansion around a point coincides with that of the original function up to a given order, but can represent the original function more efficiently and globally.

Padé approximants have been used in theoretical physics and applied mathematics for summing divergent series[25] with applications in cosmology[26], quantum electrodynamics[27], fluid dynamics[28,29] and solving the Helmholtz equation[30]. Closest in spirit to our present work is the use of Padé approximants for center manifold reduction[31,32]. The latter use is, however, restricted to equation-driven model reduction near Hopf bifurcations.

In the context of SSM-reduced models, we must consider generalizations of Padé approximants to account for multivariate functions describing the parametrization and the reduced dynamics. Since two-dimensional non-resonant SSMs are typical in applications, we focus on the bivariate and univariate cases.

In "Results", we present our results on using Padé approximants to construct reduced models on global SSMs (gSSMs). We discuss four examples, including the Kolmogorov flow[33], a nonlinear von Kármán beam in periodic and chaotic regimes[11,17], and a data-driven model of an inverted flag experiment[18]. The mathematical details of SSMs and Padé approximants are discussed in "Methods". The Supplementary Information contains further applications and examples.

## Results

### Spectral submanifolds and Padé approximants

We consider an $n$ − dimensional nonlinear dynamical system

$$
\begin{aligned}
\dot{\mathbf{x}} &= \mathbf{A}\mathbf{x} + \mathbf{f}(\mathbf{x}) + \varepsilon\mathbf{f}_{\text{ext}}(\mathbf{x}, t), \\
\mathbf{x} &\in \mathbb{R}^n, \quad \mathbf{A} \in \mathbb{R}^{n \times n}, \quad \mathbf{f} \in \mathcal{O}(|\mathbf{x}|^2), \\
n &\geq 1, \quad 0 \leq \varepsilon \ll 1
\end{aligned}
\tag{1}
$$

where $\mathbf{f}_{\text{ext}}(\mathbf{x}, t) = \mathbf{f}_{\text{ext}}(\mathbf{x}, t + T)$ represents time-periodic external forcing. We assume that the nonlinearity $\mathbf{f}$ and the forcing $\mathbf{f}_{\text{ext}}(\mathbf{x}, t)$ are also analytic functions of $\mathbf{x}$. We assume that, for $\varepsilon = 0$, the fixed point $\mathbf{x} = 0$ is hyperbolic and the spectrum of $\mathbf{A}$ is non-resonant. The slow spectral subspace is denoted $E$, and is defined as the $d$-dimensional real subspace spanned by the eigenvectors of $\mathbf{A}$ associated to its $d$ eigenvalues closest to zero and hence is an attracting, slow invariant manifold for the linearized dynamics.

We focus here on SSMs, which are the nonlinear continuations of spectral subspaces under the addition of the nonlinear terms in (1). Although there are multiple invariant manifolds tangent to the spectral subspace $E$, there is a unique smoothest one, called the primary spectral submanifold[9] denoted $\mathcal{W}(E)$. If the fixed point $\mathbf{x} = 0$ is stable, then $\mathcal{W}(E)$ is even known to be analytic. Due to the slowness of the subspace $E$, the SSM, $\mathcal{W}(E)$, is an attracting slow manifold for the autonomous system (1). The members of the invariant manifold family with reduced smoothness are called fractional (or secondary) SSMs[10]. For completeness, the necessary assumptions for the existence and uniqueness of primary SSMs[9,10,34] are recalled in "Spectral submanifolds".

In the autonomous limit with $\varepsilon = 0$, the $d$ − dimensional ($d \leq n$) primary SSM, $\mathcal{W}(E)$, can locally be represented as the image of a parametrization $\mathbf{W} : U \subset \mathbb{R}^d \to \mathbb{R}^n$, over some open set $U \subset \mathbb{R}^d$ as

$$
\mathcal{W}(E) = \left\{ \mathbf{x} = \mathbf{W}(\mathbf{p}) \mid \mathbf{p} \in U \right\} \subset \mathbb{R}^n.
\tag{2}
$$

The reduced dynamics $\dot{\mathbf{p}} = \mathbf{R}(\mathbf{p})$, with $\mathbf{R} : U \to \mathbb{R}^d$ are conjugate to (1), i.e., $\mathcal{W}(E)$ is invariant under the time evolution of (1) and therefore satisfies the invariance equation

$$
\mathbf{A}\mathbf{W}(\mathbf{p}) + \mathbf{f}(\mathbf{W}(\mathbf{p})) = D\mathbf{W}(\mathbf{p})\dot{\mathbf{p}}.
\tag{3}
$$

We refer to "Spectral submanifolds" for a discussion on SSMs of the non-autonomous system with $\varepsilon > 0$.

We solve Eq. (3) by representing the parametrization of $\mathcal{W}(E)$ and its reduced dynamics as a power series truncated to some order $N$, i.e.,

$$
\begin{aligned}
\mathbf{W}^N(\mathbf{p}) &= \sum_{|\mathbf{k}| = 0}^{N} \mathbf{W_k}\mathbf{p^k}, \\
\mathbf{R}^N(\mathbf{p}) &= \sum_{|\mathbf{k}| = 0}^{N} \mathbf{R_k}\mathbf{p^k}.
\end{aligned}
\tag{4}
$$

We define the multi-index $\mathbf{k} = (k_1, \ldots, k_d)$ and $|\mathbf{k}| = k_1 + k_2 + \ldots + k_d$, so that $\mathbf{p^k} = p_1^{k_1} p_2^{k_2} \ldots p_d^{k_d}$ refers to a scalar monomial of the components of $\mathbf{p}$ with total order $|\mathbf{k}|$. The coefficients $\mathbf{W_k}$ and $\mathbf{R_k}$ are vectors in $\mathbb{R}^n$ and $\mathbb{R}^d$, respectively, for all $\mathbf{k}$.

The coefficients $\mathbf{R_k}$ depend on the style of parametrization used. In the graph-style parametrization, the reduced coordinates are obtained as projections onto the spectral subspace $E$, while in the normal form style parametrization, non-resonant terms are set to zero. The difference between these two choices is explained in more detail by, e.g., refs. 11,22,35.

Since the primary autonomous SSM is analytic, there is a domain of convergence denoted as $U_0 \subset U \subset \mathbb{C}^d$, for which the $N \to \infty$ limit exists, i.e.,

$$
\lim_{N \to \infty} \mathbf{W}^N(\mathbf{p}) = \mathbf{W}(\mathbf{p}), \quad \forall \mathbf{p} \in U_0.
\tag{5}
$$

For a system whose slowest mode is oscillatory and not in resonance with higher modes, the optimal model reduction is achieved by a two-dimensional SSM tangent to a single oscillatory eigenspace of the autonomous problem. In that case, we can select $\mathbf{p} = (p, \bar{p})^T$ with $p \in \mathbb{C}$. With the normal form style parametrization, the reduced dynamics only contain near-resonant terms of the form $p^{k+1}\bar{p}^k$ and $p^k\bar{p}^{k+1}$, and it is conveniently expressed in polar coordinates[11]. We introduce

$$
p = \rho e^{i\theta}, \quad \bar{p} = \rho e^{-i\theta},
\tag{6}
$$

which allows us to write the SSM-reduced dynamics as

$$
\begin{aligned}
\dot{\rho} &= \kappa(\rho)\rho, \\
\dot{\theta} &= \omega(\rho).
\end{aligned}
\tag{7}
$$

The functions $\kappa(\rho)$ and $\omega(\rho)$ represent the instantaneous dependence of the damping rate on the amplitude and the frequency on the amplitude, respectively. These functions are obtained from the coefficients of the autonomous reduced dynamics, $\mathbf{R_k}$. Their Taylor expansions are of the form

$$
\kappa(\rho) = \sum_{k=0}^{\infty} \kappa_k \rho^{2k}, \quad \omega(\rho) = \sum_{k=0}^{\infty} \omega_k \rho^{2k}.
\tag{8}
$$

The software package `SSMTool`[11,36] returns the Taylor coefficients $\mathbf{W_k}$, $\kappa_k$ and $\omega_k$ up to arbitrary orders. However, the expansions (8) only

converge as long as the amplitude $\rho$ corresponding to the physical response of the system is inside the domain of convergence.

For the expansions in (8), the domain of convergence is the interior of a disk of radius $R$, the radius of convergence. As a corollary of the analyticity of holomorphic functions[23], that disk contains the nearest singularity of the complex extension of the function. This result also generalizes to the multivariate setting, although defining the domain of convergence is more complicated[37]. In addition, in contrast to the scalar case, singularities of multivariate functions are never isolated.

The a priori unknown domain of convergence of their local Taylor expansions represents a limitation of SSM-reduction approaches. Although solutions of the invariance equation (21) for SSMs exist up to any order $N$, their formal sum $\mathbf{W}^N$ has little to do with the primary SSM outside the domain of convergence $U_0$, even if $\mathbf{W}$ is well-defined on $U \backslash U_0$.

This fundamental limitation has forced most invariant manifold studies to focus on deriving reduced-order models under small perturbations near a fixed point. This, however, impedes predicting the system's response to large perturbations or the discovery of steady states far away from the known fixed points.

However, SSMs, as invariant manifolds, are known to extend globally in the phase space, as long as the flow map of the dynamical system (1) remains well-defined on them for all times. A straightforward extension of the locally known parametrization is to evolve a set of initial conditions from inside the domain of convergence globally under the flow map. This, however, assumes that the flow map of the full high-dimensional system is known in detail, making the reduced-order model redundant. This envisioned globalization is only partly achieved by the data-driven construction of SSMs[13], which starts from a limited number of trajectories of (1) and finds the observed invariant manifold using regression.

In this work, we propose a different approach to extend the range of applicability of SSM-reduced models. As long as an analytic function is known on some open domain, fundamental results in complex analysis guarantee that this knowledge can be extended to the entire domain of analyticity of the function, possibly using a different representation of the function. Switching to such a representation, other than the Taylor series (4), is known as analytic continuation[23], a powerful technique to globalize the maps $\mathbf{W}(\mathbf{p})$ and $\mathbf{R}(\mathbf{p})$.

### Globalization of invariant manifolds via Padé approximation

A well-known method for analytic continuation is the Padé approximation[24], which has had numerous applications in theoretical physics and engineering. To describe this procedure, we introduce a multivariate rational function of the form, using the same notation as in Eq. (4),

$$[N/M](\mathbf{z}) = \frac{\sum_{|\mathbf{k}|=0}^{N} a_{\mathbf{k}} \mathbf{z}^{\mathbf{k}}}{\sum_{|\mathbf{k}|=0}^{M} b_{\mathbf{k}} \mathbf{z}^{\mathbf{k}}}, \quad b_{\mathbf{0}} = 1, \quad \mathbf{z} \in \mathbb{R}^{\ell}, \tag{9}$$

where the orders of the numerator and denominator are fixed constants $N, M$. This formulation covers the cases of univariate ($\ell = 1$) and multivariate ($\ell \geq 2$) functions as well.

We represent solutions of the invariance equation (21) as rational functions of the form (9), which is achieved by requiring that the Taylor series of (9) matches that of the function to be approximated. Rational functions of the form (9) have the advantage that they can incorporate singularities that would otherwise limit the convergence of Taylor series. Based on these observations, we seek to extend SSM-reduced models using Padé approximants. We call the extended representations of an SSM the gSSM.

Increasing the orders of the numerator and the denominator in (9), one expects that the approximants provide better approximations for the functions $\mathbf{W}(\mathbf{p})$, $\mathbf{R}(\mathbf{p})$, $\kappa(\rho)$, and $\omega(\rho)$. In most practical applications, diagonal approximants (i.e., those with numerators and denominators of the same order) have proven to be the most effective. Moreover, diagonal Padé approximants of a univariate function are related to the continued-fraction representation of the function.

For meromorphic functions with an a priori unknown number of poles, strict convergence is only guaranteed in measure by the theorems of refs. 38 and [39]. These theorems cover the case of diagonal approximants and state that the sequence $[M/M](z)$ converges to the given function as $M \to \infty$ on bounded compact subsets of $\mathbb{C}$, except for sets of measure zero. Similar theorems also exist for the multivariate case[40]. These exceptional sets correspond to zero sets of the denominator in (9).

### Data-driven global reduced-order models

If the equations of motion of a dynamical system are known, we will rely on the established theory of Padé approximants and invariant manifolds and use highly optimized computational methods to solve the invariance equation (21) and construct the gSSM-reduced models.

In many practical applications, however, the governing equations are only partially known or completely unknown, and yet a predictive reduced-order model is required. In such cases, we must rely directly on data-driven methods to approximate the gSSMs. In particular, the `SSMLearn` algorithm of ref. 13 works on observations of trajectories to approximate the slow SSM $\mathcal{W}(E)$ locally.

A common use case is when a single scalar observation $y(t) \in \mathbb{R}$ is recorded. In that case, by the Takens embedding theorem[41], a $d$ − dimensional attracting SSM can be embedded generically in the space of delayed measurements,

$$\mathbf{y}(t) = (y(t), y(t - \tau), \ldots, y(t - (p-1)\tau))^T, \tag{10}$$

for some time-lag $\tau$ as long as the number of delays $p$ is more than twice the dimension of the SSM $d$. The slow SSM can then be parametrized as a graph over its tangent space at the fixed point from which it emanates. The reduced coordinates $\boldsymbol{\eta}$ on the SSM are obtained by projecting the delay-embedded measurements onto that $d$ − dimensional tangent space, i.e., letting

$$\boldsymbol{\eta} = \mathbf{V}^T \mathbf{y}, \tag{11}$$

where the columns of $\mathbf{V}$ span the tangent space. The tangent space can be efficiently approximated using the leading principal components of the delay-embedded measurements after discarding initial transients[18,19].

The parametrization of the SSM is obtained by regression using the observed trajectories and their reduced coordinates. The most straightforward choice is a polynomial basis, which is justified by the existence of a locally convergent Taylor expansion of the SSM and by the universal approximation property of polynomials[23].

Motivated by the success of Padé approximants in enhancing the convergence properties of equation-driven models, we generalize the regression step of the data-driven SSM-reduction to rational approximants. Related approaches are rational interpolation and multi-point Padé approximation. While the former requires the approximant to fit the data exactly, the latter matches the asymptotic behavior of the function at multiple expansion points.

Rational function regression[42] seeks a rational approximant of the form (9). In addition, we enforce that all components of the vector function share the same denominator and that the denominator is never zero on the training data. Having a common denominator for all components of the vector function makes it simpler to avoid spurious singularities. We elaborate on the steps of this regression task in "Rational function regression".

## Example 1: connecting orbit in Kolmogorov flow

Our first example is the 2D Kolmogorov flow, governed by the Navier-Stokes equations in a periodic domain subject to spatially periodic forcing. The flow domain is given by $x, y \in [0, 2\pi]$ and the time evolution of the vorticity $\omega(x, y) = \nabla \times \mathbf{u}$ is governed by the non-dimensionalized equation

$$\frac{\partial \omega}{\partial t} = -(\mathbf{u} \cdot \nabla)\omega + \frac{1}{Re}\Delta\omega - L\cos Ly, \tag{12}$$

where $L$ denotes the forcing wave number. This influences the size of the large-scale flow structures, the bifurcations observed in (12) and the properties of the turbulent dynamics at high $Re$[43]. The laminar solution is a fixed point of (12) for all Reynolds numbers that can be written as

$$\omega_0(x, y) = -\frac{Re}{L}\cos Ly. \tag{13}$$

Following[33], we set $L = 4$ and discretize the system using $576 = 24 \times 24$ Fourier modes. The numerical implementation is based on ref. 44 and results in a system of ODEs in the form (1) for the Fourier amplitudes $\hat{\omega}(k_x, k_y)$ with $n = 576$. Further details on the implementation can be found in the Supplementary Information.

The laminar flow $\omega_0$ is already unstable for low Reynolds numbers, losing stability in a bifurcation around $Re = 9.1$. The state $\omega_0$ is connected to two new stable fixed points $\omega_{1,2}$ by its 1D unstable manifold. This unstable manifold coincides with the 1D slow SSMs of $\omega_{1,2}$, forming two heteroclinic orbits.

We fix $Re = 11$ and consider the stable fixed point $\omega_1$. We compute the parametrized slow SSM of $\omega_1$ as i.e., $\hat{\omega} = \mathbf{W}(\xi)$. This computation is carried out automatically by `SSMTool`, which returns the coefficients of the Taylor expansion of $\mathbf{W}(\xi)$ and the reduced dynamics on the SSM as

$$\hat{\omega}(k_x, k_y) = \mathbf{W}(\xi) = \sum_{k=0}^{N}\mathbf{W}_k\xi^k, \tag{14}$$

$$\dot{\xi} = R(\xi) = \sum_{k=0}^{N}R_k\xi^k. \tag{15}$$

We visualize the connecting orbit, i.e., the 1D slow SSM in a 3D slice of the 576-D phase space in Fig. 1a. The stable fixed points $\omega_{1,2}$ are marked with black and blue dots, and the unstable fixed point $\omega_0$ is red. The heteroclinic orbit, which is obtained by direct numerical integration of (12) (black curve), is approximated by the Taylor expansion of the SSM up to order-16. The domain of convergence is clearly limited, and it does not contain the unstable fixed point.

To construct a globalized slow SSM, we compute the Padé approximant of each component (Fourier mode) of the vector $\mathbf{W}(\xi)$. Although, in principle, spurious poles could arise for each component separately, we find that the diagonal approximants are well-behaved. An alternative approach would be to construct the vector Padé approximants[45], that is, to find a common denominator for all components of $\mathbf{W}(\xi)$[46].

The componentwise computed $[5/5](\xi)$ Padé approximant approximates the heteroclinic orbit connecting $\omega_1$ and $\omega_0$ remarkably well, as shown by the orange curves in Fig. 1a. We also observe that the manifold $\mathcal{W}(E)$ can only be represented as a graph over $E$ for this segment, since the derivative $\frac{\partial}{\partial\xi}\mathbf{W}(\xi)$ diverges at a fold point near $\omega_0$. The parametrization, therefore, cannot be continued to capture $\omega_2$. Note, however, that the Taylor approximation diverges well before encountering this unremovable singularity of the graph-style parametrization.

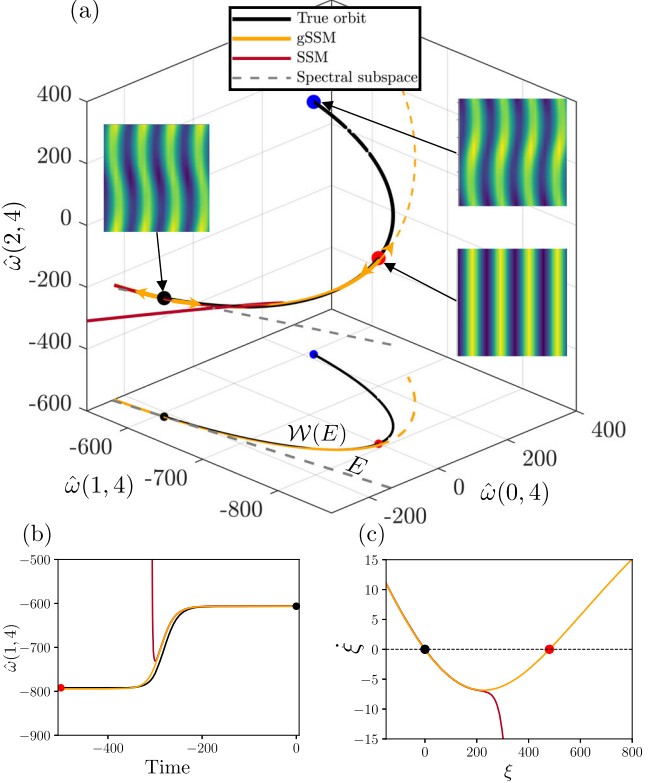

**Fig. 1 | Model reduction for the Kolmogorov flow.** Heteroclinic orbits (black) connect $\omega_{1,2}$ and $\omega_0$. **a** Projection of the phase space onto three dominant Fourier modes (1, 4), (0, 4) and (2, 4). We show the slow SSM $\mathcal{W}(E)$ (black), which is tangent to the spectral subspace $E$ (gray), its order-16 Taylor expansion (red) near the fixed point $\omega_1$, and the order [5/5] gSSM approximation (orange). The curves are also projected onto the horizontal axes. The vorticity fields corresponding to the three fixed points are shown in the insets. **b** A trajectory on the heteroclinic orbit obtained by backwards integration and its SSM-reduced and gSSM-reduced counterparts. **c** SSM-reduced and gSSM-reduced dynamics.

To verify the validity of the reduced-order models, predictions should be compared to trajectories of the full system. However, since the fixed point is stable, a nearby initial condition will leave its neighbourhood along the heteroclinic orbit only in backward time. Therefore, we integrate the initial condition $\xi(0) = 10^{-5}$, close to the stable fixed point, backward under the SSM-reduced and the gSSM-reduced dynamics.

Figure 1b shows the SSM-reduced trajectory, which exhibits finite-time blowup and is only a reliable model near the stable fixed point. In contrast, the gSSM-reduced trajectory converges to $\omega_0$ in backward time and hence captures the global dynamics accurately.

Figure 1c shows the reduced dynamics $\dot{\xi}$ on the SSM and the gSSM. In addition to the trivial fixed point at $\xi = 0$, the gSSM-reduced model contains the unstable fixed point $\omega_0$, given by the intersection with $\dot{\xi} = 0$ and hence provides a robust reduced model of the system. In contrast, based on the Taylor approximated SSM, the existence of an unstable fixed point cannot be concluded.

## Example 2: Von Kármán beam

We now consider the model of a nonlinear von Kármán beam with clamped-free boundary conditions[47], shown in Fig. 2a. The beam has length $L = 1$ m, height $h = 1$ mm, width $b = 0.1$ m, Young's modulus $E = 70$ GPA, viscous damping rate $\alpha = 10^7$ Pa s and density 2700 kg/m$^3$. We use a finite element discretization with 10 elements, resulting in 30 degrees of freedom, with a phase space of dimension $n = 60$. The discretization is implemented in the `SSMTool`[36] repository.

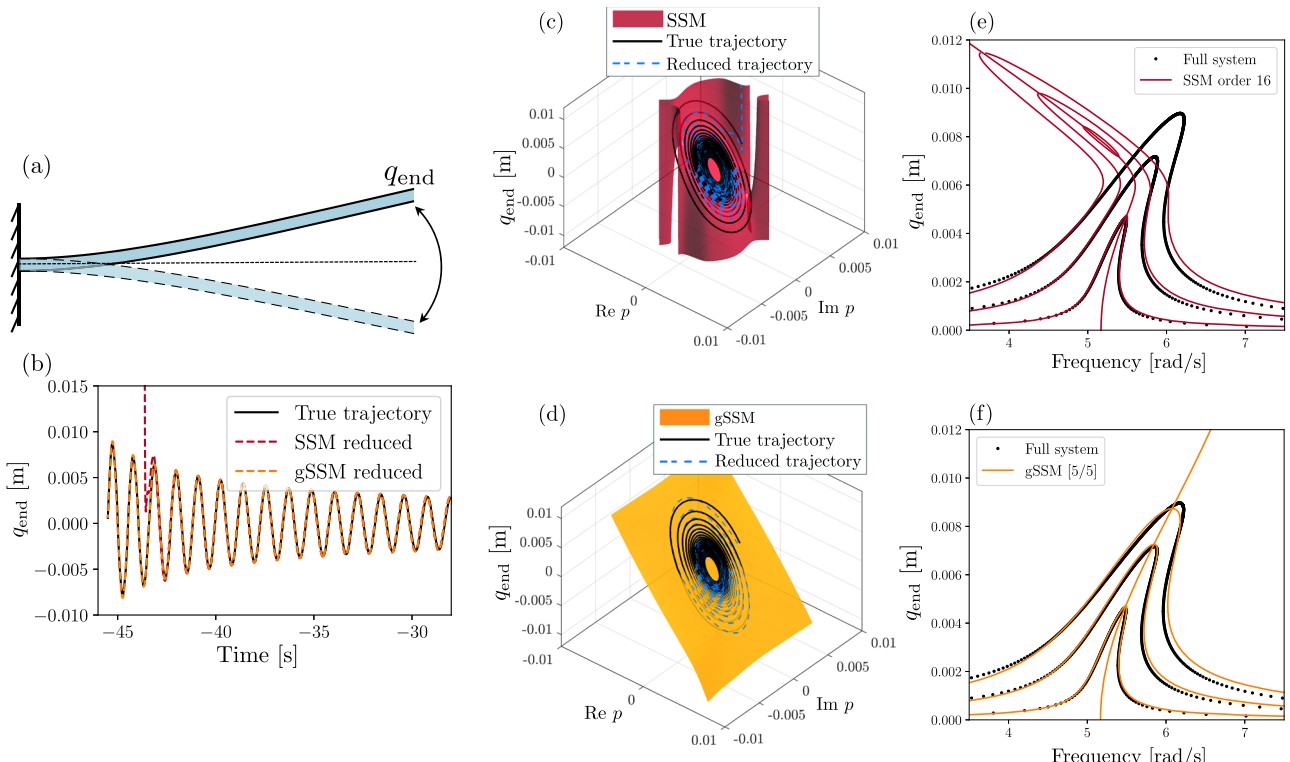

**Fig. 2 | Model reduction of a von Kármán beam. a** von Kármán beam with clamped-free boundary conditions. **b** Trajectory of the unforced system (black) with its order-16 SSM approximation (red) and gSSM-approximation (orange) with a [5/5] and [5/4] Padé approximant. **c**, **d** Representation of the end point displacement $q_{\text{end}}$ and the full-order trajectory shown in (**b**) with the SSM and gSSM-reduced trajectories. **e**, **f** Forced response defined as the maximal end point displacement due to a forcing amplitude $\varepsilon = 0.5, 1.7, 2.8$. **e** Shows the SSM-prediction and (**f**) shows the gSSM prediction. A supplementary animation showing the gSSM-prediction is also available.

We construct the slowest SSM of the undeformed configuration, which is tangent to the spectral subspace corresponding to the eigenvalues $\lambda_{1,2} = -0.0019 \pm 5.1681i$. Due to the Taylor approximation, the autonomous SSM can only capture the decaying oscillations of the beam up to amplitudes of around 5 mm, which is verified by backward integration of the SSM-reduced and the full models. This can be seen in Fig. 2b–c, which shows the autonomous trajectory exiting the domain of convergence of the Taylor series. This is evident for the high-order Taylor approximant, which exhibits finite-time blowup.

In contrast, the gSSM model, globalized using a [5/5] Padé approximant for the reduced dynamics and a [5/4] approximant for the parametrization, remains well-behaved for even larger amplitudes. Due to spurious singularities in the parametrization, we chose the [5/4] approximant instead of [5/5]. Since the convergence of Padé-approximants is only guaranteed in measure, singularities coinciding with zero sets of the denominator of (9) must be actively avoided before deploying the reduced-order model, as we demonstrate in the Supplementary Information.

The backbone curve, i.e., the instantaneous relationship between the normal form amplitude $\rho$ and the frequency, is given by $\omega(\rho)$ in (7). The Taylor expansion of $\omega(\rho)$ and its [5/5] Padé approximant are given as

$$
\begin{aligned}
\omega(\rho) &= 5.16 + 9.3 \cdot 10^4 \rho^2 - 4.5 \cdot 10^9 \rho^4 \\
&\quad + 4.2 \cdot 10^{14} \rho^6 + O(\rho^8), \\
[5/5](\rho) &= \frac{5.16 + 10^6 \rho^2 + 3.7 \cdot 10^{10} \rho^4}{1 + 1.8 \cdot 10^5 \rho^2 + 4.7 \cdot \rho^4}.
\end{aligned}
\tag{16}
$$

The Taylor coefficients in (16) are growing rapidly, and the alternating sign pattern of the coefficients suggests the convergence-limiting singularity is along the imaginary axis. This can be inferred

using the method of refs. 48,49, which we specialize to our examples in the Supplementary Information.

When an external forcing $\varepsilon \mathbf{f}_{ext} \cos(\Omega t)$ is applied to the beam with a frequency $\Omega$ almost in resonance with the slowest eigenfrequency $\mathrm{Im}(\lambda_{1,2})$, we can use the unforced SSM to make predictions about the forced response. To leading order, the reduced dynamics of the forced system are simply a perturbed version of those of the unforced one (see "Spectral submanifolds"). Furthermore, the forced response corresponds to the periodic orbits of the reduced model. For 2D SSMs, these are directly given by the equation (26) in "Spectral submanifolds".

We compare the SSM-reduced forced response to the forced response of the full system, obtained by direct numerical continuation using COCO[50]. Predictions with a high-order Taylor approximation of the SSM are shown in Fig. 2e. They accurately capture the forced response for small forcing amplitudes up to $\varepsilon = 0.5$, as initially reported by ref. 11. For higher amplitudes, the Taylor approximated reduced trajectory reaches its boundary of convergence, and the model breaks down. In contrast, the gSSM prediction using the [5/5] Padé approximant remains accurate for larger amplitudes as well. Note that the small errors in the peaks of the forced response curve are due to our initial assumption of a small forcing amplitude. Indeed, (26) technically holds only for small $\varepsilon$.

### Example 3: chaotic von Kármán beam

In our next example, we construct a 2D mixed-mode SSM to characterize the chaotic behavior of a periodically forced buckled von Kármán beam, shown in Fig. 3. We adopt the same finite element code used in "Example 2: Von Kármán beam", developed by ref. 11, with pinned-pinned boundary conditions. To induce buckling, a compressive force is applied at the rightmost element, equal to 145% of the critical value $f_{crit} = 1.5$ kN. This gives rise to a pair of stable fixed points.

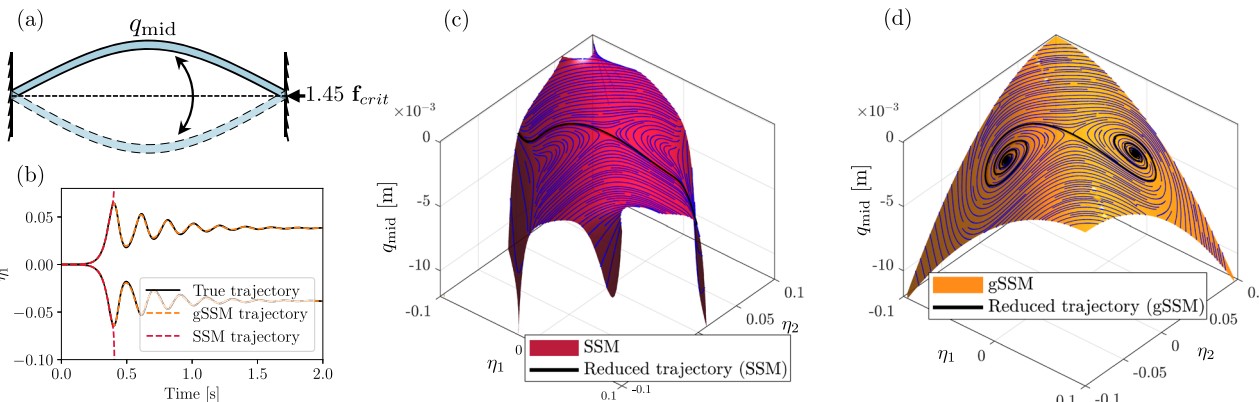

**Fig. 3 | Reduced model of the buckled beam. a** Sketch of the beam in the buckled configuration, with no external forcing. **b** Trajectories in the unstable manifold of the unstable fixed point (black). Their order-18 SSM-reduced (red) and order [6/6] gSSM-reduced (orange) approximations are also shown. **c, d** The SSM and gSSM in the physical space with the direction field of the reduced dynamics indicated on the surface of the manifold. The predicted trajectories connecting the unstable fixed point to the stable fixed points are shown in black.

The same system, much closer to the critical point, was also analyzed using a data-driven model by ref. 17.

The spectral subspace of the linear part associated with the buckling instability is two-dimensional and has the corresponding eigenvalues $\lambda_1 = 23.48$ and $\lambda_2 = -23.48$. The other modes are all stable and correspond to oscillatory dynamics. This eigenvalue configuration indicates a considerable departure from criticality and center manifold theory.

We use `SSMTool` to find an order-18 approximation of the 2D SSM of the unforced beam and its reduced dynamics. Since the SSM is tangent to a spectral subspace with real eigenvalues, we denote the reduced coordinates as $\boldsymbol{\eta} = (\eta_1, \eta_2) \in \mathbb{R}^2$. The reduced dynamics are given by

$$
\begin{aligned}
\dot{\eta}_1 &= 23.48\eta_1 - 2800\eta_1^3 - 1760\eta_2^3 - 7297\eta_1^2\eta_2 \\
&\quad - 6268\eta_1\eta_2^2 + O(|\eta|^4), \\
\dot{\eta}_2 &= -23.52\eta_2 + 2790\eta_1^3 + 1760\eta_2^3 + 7297\eta_1^2\eta_2 \\
&\quad + 6268\eta_1\eta_2^2 + O(|\eta|^4).
\end{aligned}
\tag{17}
$$

Analysis of the reduced dynamics shows, however, that the fixed points born due to the buckling instability lie outside the domain of convergence. We show in Fig. 3b the time series of a trajectory initialized on the unstable manifold of the fixed point. The SSM-reduced trajectory based on Taylor expansion blows up once the reduced trajectory exits the domain of convergence. In contrast, a comparable, [6/6] Padé approximant and the gSSM-reduced trajectory correctly capture the convergence to both of the buckled states.

This is even more apparent in Fig. 3c, d, showing the image of the parametrization and the direction field of the reduced dynamics. The SSM-reduced model in Fig. 3b predicts diverging, unphysical displacements for the mid-point of the beam. The gSSM-model in Fig. 3c correctly identifies all fixed points and the orbits connecting them. We have, therefore, extended the SSM-reduced model obtained from local information around the unstable fixed point to a globally valid one.

We now extend the autonomous gSSM model to account for periodic forcing. A periodic force on the middle node of the beam, as shown in Fig. 4a, can make the dynamics of the beam chaotic. This has been observed and reported in the data-driven model of ref. 17, who used simulation data of the full system. We now characterize the chaotic behavior without relying on any full-order simulations.

To compare the full-order and the reduced-order dynamics, we take one of the buckled fixed points as an initial condition and simulate the beam under the influence of periodic forcing acting on the mid-

point with $\Omega = 25.3$ rad/$s$ and $|\varepsilon\mathbf{f}_{ext}| = 21.1$ N. Since this initial condition is known to be on the SSM, we run the same trajectory with the reduced dynamics after adding the leading-order contribution of the forcing as in (23).

The full-order trajectory is shown in Fig. 4b. As expected from the previous analysis and Fig. 3, the forced SSM-reduced model blows up within a fraction of a second. In contrast, the globally valid gSSM model exhibits sustained chaotic behavior, closely matching the full-order trajectory for short times. Figure 4d–e shows that the chaotic attractor appearing as a result of the periodic forcing extends way outside the domain of convergence of the Taylor series of the SSM.

We also construct the Poincaré-map of the full model and the gSSM model by sampling the trajectories at multiples of the driving period $T = \frac{2\pi}{\Omega}$. Because the gSSM-model is a simple 2D ODE, we can sample the Poincaré-map with a fine resolution to obtain the structure in the reduced phase space shown in Fig. 4c. Overlaying the Poincaré-map obtained from the full system, we see a close correspondence with the predicted attractor.

In addition, we estimate the leading Lyapunov exponents based on the exponential rate of divergence of initially close trajectories[51] as $\lambda_{gSSM} = (3.0 \pm 0.02)1/s$ and $\lambda_{full} = (3.1 \pm 0.05)1/s$. The reduced model is in close agreement with the full model, even though the forced dynamics were approximated by simply projecting the forcing term onto the tangent space of the SSM according to (23). Incorporating higher-order corrections of the forced dynamics, as in (22), further improves the model, leading to more accurate short-time predictions. We present these comparisons, along with additional properties of the chaotic gSSM model, in the Supplementary Information.

## Example 4: data-driven model of an inverted flag experiment

We now consider, as a data-driven example, the dynamics of an inverted flag, which is a flexible elastic sheet in the counterflow of a water tunnel. This configuration has generated recent interest due to its applications in energy harvesting[52] and vegetation[53]. From a modeling perspective, the inverted flag is a complicated fluid-structure interaction problem that benefits from reduced-order modeling. Here, we rely on experimental data obtained by ref. 18, whose experimental configuration is shown in Fig. 5a–b. The elastic sheet is mounted in a water tunnel and is recorded from below with a video camera. Using classical image processing tools, the deflection of the tip of the flag $y(t)$ is recorded during the experiment.

The main parameters of the system are the bending stiffness $K_B$ and the Reynolds number of the incoming flow, governed by the mean velocity $U$. By tracking the displacement of the tip of the flag, a data-

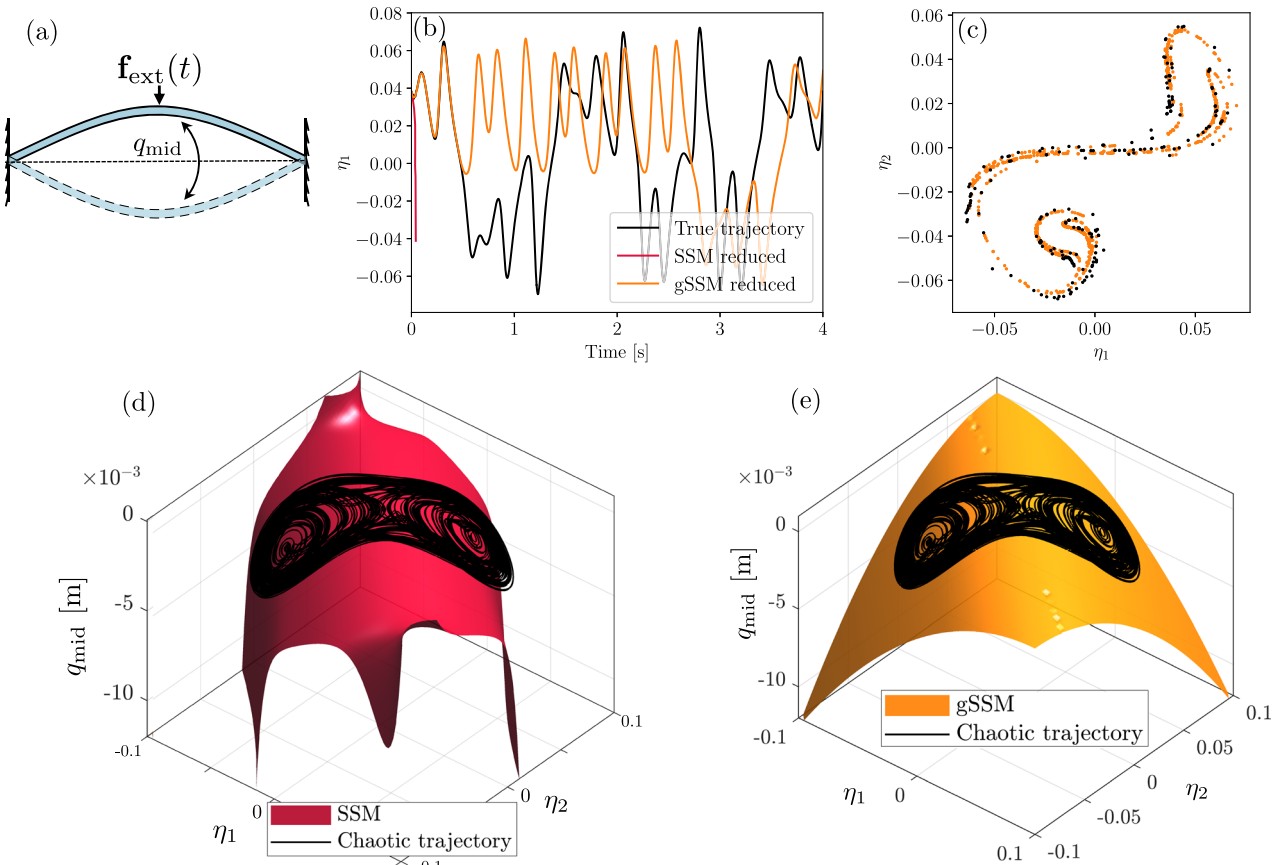

**Fig. 4 | Chaotic response of the buckled beam. a** Buckled von Kármán beam with periodic external forcing. **b** Time series of the reduced coordinate $\eta_1$ on a chaotic trajectory of the full system (black). Also shown are the SSM-reduced forced model, which diverges immediately (red), and the gSSM-reduced trajectory (orange). **c** Sampling of the Poincaré map of the gSSM-reduced model (orange) and the true system (black). **d**, **e** The autonomous SSM and gSSM with the chaotic trajectory of the full model. A supplementary animation comparing the SSM- and gSSM-predictions is also available.

driven SSM-reduced model was obtained[18] for both the large-amplitude periodic flapping regime and the chaotic flapping regime. In these regimes, the undeflected state of the flag is an unstable fixed point, which has a low-dimensional attracting mixed-mode SSM.

We focus here on the large-amplitude flapping regime with $K_B = 0.21$ and $\mathrm{Re} = 6 \times 10^4$. The slow SSM is tangent to a spectral subspace associated with an unstable real eigenvalue and a stable real eigenvalue, and is two-dimensional. In addition to the saddle-type fixed point corresponding to the undeflected flag, the slow SSM contains two additional fixed points, which correspond to the deflected, but still stationary flag. The periodic flapping is a stable limit cycle within this slow SSM.

We now construct a data-driven gSSM-reduced model based on the tip-deflection data of ref. 18. A total of 16 trajectories are used for training. To reconstruct the slow SSM, we embed the tip displacement data $y(t)$ trajectories using $p = 25$ time delays to form the observable $\mathbf{y} \in \mathbb{R}^{25}$. As in ref. 18, we approximate the tangent space of the SSM at the fixed point using the two leading principal components of the delay-embedded trajectories. Since we use a moderate number of time delays and a short delay time, a leading-order (linear) approximation for the SSM suffices, i.e., we let

$$\begin{pmatrix} \eta_1 \\ \eta_2 \end{pmatrix} = \boldsymbol{\eta} = \mathbf{V}^T \mathbf{y}, \quad \mathbf{y} = \mathbf{V}\boldsymbol{\eta}. \tag{18}$$

The reduced dynamics are now approximated using rational functions, as detailed in "Rational function regression", by solving the minimization problem (44). We find that a [5/5] approximant gives the

optimal reconstruction error, as computed on a validation trajectory, which was not used in the training. The resulting reduced vector field and the predictions on test trajectories are shown in Fig. 5c–e. A diagonal approximant also ensures that the reduced dynamics remain well-behaved outside the domain of the training data. As opposed to classical polynomial regression, the values of the approximants remain bounded or only grow mildly for large $|\boldsymbol{\eta}|$.

To obtain a comparable test accuracy to that shown in Fig. 5d–e, previously, an order-11 polynomial approximation was used for the reduced dynamics. This required determining a total of 154 coefficients $\mathbf{R_k}$. The data-driven rational approximation, on the other hand, requires only 60 coefficients, which is a significant reduction, for the cost of a slightly increased computational burden. In addition, the rational approximants extrapolate to larger domains in a more controlled way.

As we show in the Supplementary Information, preventing the rational functions from becoming singular is essential for an accurate approximation. Other available methods, such as the rational function extension of the Sparse Identification of Nonlinear Dynamics (SINDy) algorithm[54,55] enforce no such constraints.

## Discussion

We have presented a method to extend the range of validity of invariant manifold-based reduced-order models by applying Padé approximation, a classic analytic continuation technique. The Taylor coefficients of the parametrization of the SSM and the reduced dynamics were obtained with the robust numerical routines of SSMTool. We then extend their range of validity globally to obtain

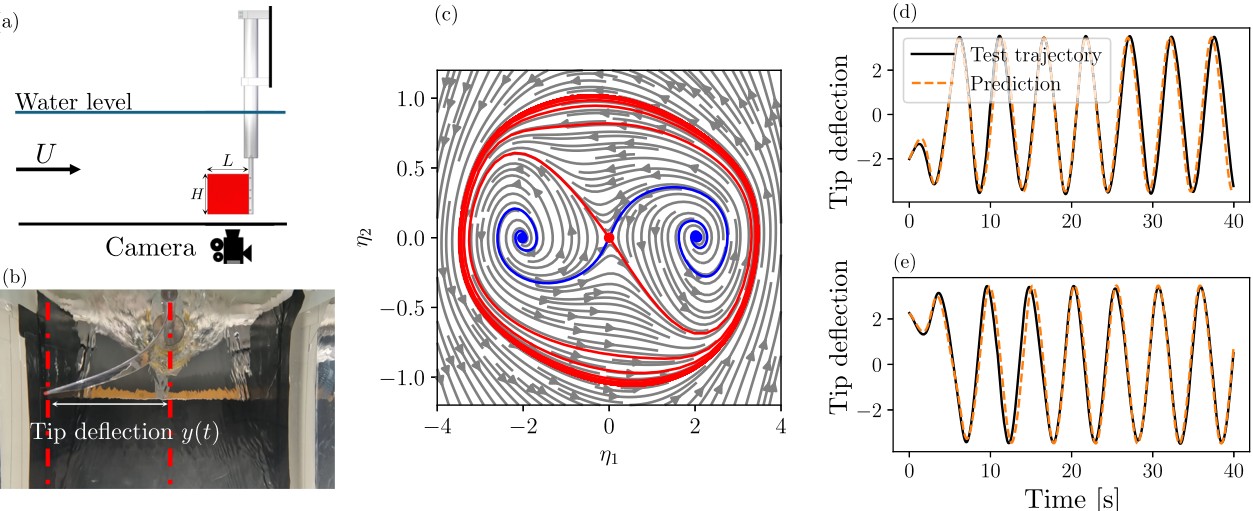

**Fig. 5 | Reduced model of the inverted flag experiment. a** Experimental setup of the inverted flag and snapshot of the experiment (**b**), courtesy of Giovanni Berti. The geometric parameters are $H = 150$ mm, $L = 150$ mm, $U = 1$ m/s. **c** Phase portrait of the gSSM-reduced dynamics of the inverted flag obtained from the $[5/5](\boldsymbol{\eta})$ approximation. The unstable fixed points are marked with colored dots. Blue curves denote the stable manifold of the undeflected state, which connects to the two coexisting deflected fixed points. The red curve denotes the unstable manifold of the saddle, which wraps around the stable limit cycle. **d, e** Predictions of the reduced model on test trajectories. The black curve is the true trajectory, and the dotted orange curve is the gSSM prediction.

the gSSM-reduced model, by applying Padé approximation to these mappings that agree with the local Taylor expansions up to a prescribed order.

We have demonstrated the method on high-dimensional examples of dynamical systems exhibiting global nonlinear behaviour, such as transitions between steady states, large-amplitude oscillations, or even chaotic behaviour. In all cases, gSSM models obtained with Padé approximation significantly extend the domain of validity of the reduced model, reaching well beyond the domain of convergence of the classical Taylor series. We have also shown that the sign pattern of the univariate Taylor coefficients can be used to infer the location of the singularity.

To apply our approach, unexpected singularities in the approximants must be checked, and the optimal approximant of the Padé table must be determined according to the steps we have outlined in "Padé approximants". Luckily, the numerical effort required to compute Padé approximants is negligible compared to the computation of the original Taylor coefficients of the SSM and the reduced dynamics. Therefore, even if no singularity-free Padé approximants can be found up to a given order, the benefits of a gSSM model far outweigh the costs of deriving it.

We have shown that a data-driven analog of Padé approximation, rational function regression, can be directly applied to experimental data. Although rational function regression requires more computational power than the standard polynomial approximation implemented in `SSMLearn`, it can produce more accurate reduced models with fewer unknown coefficients, resulting in ultimately simpler models.

We have demonstrated the method on the inverted flag example of ref. 18, where the rational approximants of the SSM and the reduced dynamics were able to predict the coexisting deflected and undeflected fixed points and the stable limit cycle. As a main benefit, we note that the data-driven approach does not suffer from the same singularity issues as the equation-driven one.

As we have demonstrated, Padé approximants and their data-driven extensions are particularly well-suited for approximating the global reduced dynamics on invariant manifolds. In addition to being able to represent more complex functions, they retain physical interpretability. In contrast to modern machine learning methods, the coefficients of the gSSM-reduced dynamics can be directly related to

the underlying physics of the system. Specifically, the coefficients of the rational function can be interpreted as nonlinear frequencies or damping rates, as we have shown in "Example 2: Von Kármán beam".

Finally, we note that although we have focused here on SSM-reduced models, the globalization method applies to other perturbative methods for dynamical systems, such as Poincaré-Lindstedt series[56], geometric singular perturbation theory[5,6], or model reduction based on local linearization results[57].

## Methods
### Spectral submanifolds

Let us assume that $\mathbf{x} = 0$ is a hyperbolic fixed point of the system (1) with $\varepsilon = 0$ and $\lambda_1, \ldots, \lambda_n \in \mathbb{C}$ are the eigenvalues of $\mathbf{A}$ with the corresponding eigenvectors denoted $\mathbf{e}_1, \ldots, \mathbf{e}_n \in \mathbb{C}^n$. We assume that a spectral gap condition holds for some $d < n$, i.e.,

$$\mathrm{Re}\,\lambda_n \leq \ldots \leq \mathrm{Re}\,\lambda_{d+1} < \mathrm{Re}\,\lambda_d \leq \ldots \leq \mathrm{Re}\,\lambda_1. \tag{19}$$

Let us denote the $d$ − dimensional slow spectral subspace of $\mathbf{A}$ as $E$, which is defined as the span of the real and imaginary parts of the eigenvectors corresponding to the $d$ eigenvalues closest to zero. If the nonresonance condition

$$\lambda_k \neq \sum_{j=1}^{n} m_j \lambda_j, \quad k = 1, \ldots, n,$$
$$m_j \in \mathbb{N}, \quad \sum_{j=1}^{n} m_j > 1, \tag{20}$$

holds, a family of SSMs exists tangent to $E$, as discussed in refs. 9,10. The primary SSM is the unique, smoothest member of the family of $d$ − dimensional invariant manifolds tangent to $E$ at the origin and is denoted as $\mathcal{W}(E)$[9,10].

SSMs also exist for the non-autonomous system, i.e., for $\varepsilon > 0$. In that case, SSMs are slow invariant manifolds attached to an anchor trajectory[34]. For simplicity, we focus here on the case of periodic forcing with a single harmonic, where the anchor trajectory is a periodic orbit, and the original formulation of ref. 11 applies. Denoting the parametrization of the slow SSM as $\mathbf{W}_\varepsilon(\mathbf{p}, \Phi)$, where $\Phi$ is the phase of

the periodic forcing, the invariance equation reads as

$$\mathbf{A}\mathbf{W}_\varepsilon(\mathbf{p}, \Phi) + \mathbf{f}(\mathbf{W}_\varepsilon(\mathbf{p}, \Phi)) + \varepsilon\mathbf{f}_{\text{ext}} \cos\Phi$$
$$= D\mathbf{W}_\varepsilon(\mathbf{p}, \Phi)\dot{\mathbf{p}} + D_\Phi\mathbf{W}_\varepsilon\dot{\Phi}, \qquad (21)$$

where $\dot{\Phi} = \Omega$ is the forcing frequency. The reduced dynamics is denoted as $\dot{\mathbf{p}} = \mathbf{R}_\varepsilon(\mathbf{p}, \Phi)$.

Due to the guaranteed smoothness of primary SSMs, the invariance equation can be solved using a Taylor expansion in $\mathbf{p}$ and a Fourier expansion in $\Phi$. The coefficients are then obtained by imposing the invariance equation order-by-order in the reduced coordinates. As shown by refs. 58–60, to first order in $\varepsilon$ and to order-$\hat{N}$ in the reduced coordinates, the resulting expression is

$$\mathbf{R}_\varepsilon(\mathbf{p}, \Phi) = \mathbf{R}(\mathbf{p}) + \varepsilon \sum_{|\mathbf{k}|=0}^{\hat{N}} \mathbf{S}_\mathbf{k}(\Phi)\mathbf{p}^\mathbf{k} + O(\varepsilon^2), \qquad (22)$$

where the coefficients $\mathbf{S}_\mathbf{k}(\Phi)$ can also be Fourier-expanded. Accounting for the phase-dependence of the nonautonomous coefficients in the reduced dynamics increases the accuracy of the reduced models for higher forcing amplitudes, at the cost of computing the coefficients $\mathbf{S}_\mathbf{k}(\Phi)$ for each forcing frequency of interest. These computations are already implemented in `SSMTool`[60].

However, the phase-dependence of the parametrization and the reduced dynamics are often only small effects. It is, therefore, common to keep only the leading-order phase-dependent term in the reduced dynamics, resulting in

$$\mathbf{R}_\varepsilon(\mathbf{p}) = \mathbf{R}(\mathbf{p}) + \varepsilon\mathbf{V}^*\mathbf{f}_{ext}\cos(\Omega t), \qquad (23)$$

where the operator $\mathbf{V}^*$ projects onto the tangent space and can be computed using the left eigenvectors of $\mathbf{A}$. Specifically, for a non-resonant two-dimensional SSM, we have

$$\dot{\rho} = \kappa(\rho)\rho + \varepsilon f \sin\psi \qquad (24)$$

$$\dot{\psi} = \omega(\rho) - \Omega + \varepsilon f \frac{1}{\rho}\cos\psi, \qquad (25)$$

where we have introduced the phase lag $\psi = \theta - \Omega t$, and the forcing amplitude $f$ is the projection of the external forcing amplitude[11].

The forced response of the system is then obtained by seeking fixed points of the reduced dynamics, which are given by the solutions of the equation $\dot{\rho} = \dot{\psi} = 0$. The amplitude $\rho^*$ of the response satisfies the implicit equation[11]

$$\left(\Omega - \omega(\rho_*)\right)^2 - \frac{\varepsilon^2 f^2}{\rho_*^2} + \kappa(\rho_*)^2 = 0. \qquad (26)$$

## Padé approximants
We now seek to improve the convergence properties of Taylor series approximations by summing the series outside the domain of convergence. One of the most popular methods of analytic continuation is Padé approximation, which is often used to sum divergent perturbative series. Padé approximation is primarily carried out on functions of a single complex variable, and hence it is directly applicable to 1D SSMs and to the functions $\kappa(\rho)$ and $\omega(\rho)$.

Consider the function

$$f(z) : \mathbb{C} \to \mathbb{C}, \qquad (27)$$

which could represent the parametrization of an invariant manifold, the reduced dynamics or $\kappa(\rho)$ and $\omega(\rho)$ in (26). Let us denote its Taylor

series representation around $z = 0$ as

$$f(z) = \sum_{k=0}^\infty c_k z^k. \qquad (28)$$

The Padé approximant of type $(N, M)$ is defined as the rational function

$$[N/M](z) = \frac{\sum_{k=0}^N a_k z^k}{\sum_{k=0}^M b_k z^k}, \qquad (29)$$

where $a_0 = f(0)$ and $b_0$ can be chosen as $b_0 = 1$ without loss of generality. The Padé approximant is the best rational approximation of $f$ around $0$[24] in the sense that its Taylor expansion around 0 is the same as that of $f$ up to order $N + M$, i.e.,

$$f(z) \sum_{k=0}^M b_k z^k - \sum_{k=0}^N a_k z^k = O\left(z^{N+M+1}\right). \qquad (30)$$

Therefore, the coefficients $b_k$ and $a_k$ can be determined by requiring

$$\sum_{k=0}^\infty c_k z^k \sum_{m=0}^M b_m z^m = \sum_{k=0}^N a_k z^k \qquad (31)$$

for orders up to $N + M$ in $z$. This results in the linear equations for the coefficients $a_k, b_k$,

$$a_k = \sum_{m=0}^k c_{k-m} b_m, \quad k = 0, 1, \dots, N + M. \qquad (32)$$

We follow the robust approach of ref. 61, who first solve the homogeneous equations

$$\sum_{m=0}^k c_{k-m} b_m = 0, \quad k = N+1, \dots, N+M, \qquad (33)$$

with the convention of $b_j = 0$ for $j < 0$ or $j > M$. Using the singular value decomposition (SVD) of the Toeplitz matrix with elements $[c_{k-m}]$, the coefficients $b_k$ can be computed. The remaining unknown coefficients $a_k$ are then given as

$$a_k = \sum_{m=0}^k c_{k-m} b_m, \quad k = 0, 1, \dots, N. \qquad (34)$$

This method is robust against numerical errors in the Taylor coefficients. In addition, many of the spurious poles of the approximant are removed, although not all of them[62].

To apply Padé approximation to higher-dimensional SSMs, we need to generalize the method to multivariate power series. The multivariate generalization is not as straightforward as the univariate case, but multiple definitions exist. The most commonly used are Chisholm approximants[63] and the homogeneous approximants[24,40,64]. Let us consider the multivariate function

$$f : \mathbb{C}^d \to \mathbb{C}. \qquad (35)$$

given as a convergent Taylor series

$$f(\mathbf{z}) = \sum_{|\mathbf{k}|=0}^\infty c_\mathbf{k} \mathbf{z}^\mathbf{k}. \qquad (36)$$

We adopt the homogeneous approximants (9), as defined originally by refs. 40,64 and require that the Taylor expansion of $[N/M](\mathbf{z})$

around $\mathbf{z} = 0$ coincides with that of $f(\mathbf{z})$ up to order $N + M$, as

$$f(\mathbf{z}) \sum_{|\mathbf{k}|=0}^{M} b_{\mathbf{k}} \mathbf{z}^{\mathbf{k}} - \sum_{|\mathbf{k}|=0}^{N} a_{\mathbf{k}} \mathbf{z}^{\mathbf{k}} = O(\mathbf{z}^{N+M+1}) \tag{37}$$

$$\sum_{|\boldsymbol{\ell}|=0}^{|\mathbf{k}|} c_{\mathbf{k}-\boldsymbol{\ell}} b_{\boldsymbol{\ell}} = a_{\mathbf{k}} \quad \text{for } |\mathbf{k}| = 0, 1, \ldots, N + M. \tag{38}$$

Since two-dimensional, oscillatory SSMs are the relevant objects for most systems, we consider the bivariate case as an illustration, i.e., with $\mathbf{z} = (z_1, z_2)^T$, where the coefficients are indexed on a lattice $\mathcal{L}_{N_1, N_2}$ defined as $\mathcal{L}_{N_1, N_2} = \{(i,j) \in \mathbb{N}^2 : N_1 \le i + j \le N_2\} \subset \mathbb{N}^2$.

The conditions (38) become

$$\sum_{k=0}^{\alpha} \sum_{\ell=0}^{k} c_{\beta-\ell, \alpha-k-\beta+\ell} b_{\ell, k-\ell} = a_{\beta, \alpha-\beta}, \tag{39}$$

for $(\alpha, \beta) \in \mathcal{L}_{0,N}$ and

$$\sum_{k=0}^{\alpha} \sum_{\ell=0}^{k} c_{\beta-\ell, \alpha-k-\beta+\ell} b_{\ell, k-\ell} = 0, \tag{40}$$

for $(\alpha, \beta) \in \mathcal{L}_{N,N+M}$. Note, however, that the total number of unknowns and equations in (39) and (40) are not equal. Therefore, we seek a least-squares solution to (40)[65].

To summarize, gSSM-reduction consists of the following steps to compute the [N/M] Padé approximants of the parametrization and the reduced dynamics of a slow SSM.

1. Compute the Taylor series expansion of the SSM up to order $N + M$ using either the normal form style parametrization or the graph-style parametrization. This returns $\mathbf{W}(\mathbf{p})$, $\omega(\rho)$ and $\kappa(\rho)$.
2. Solve the homogeneous linear equations (40) and (33) for the coefficients of the denominators of the parametrization and of $\omega$ and $\kappa$ using the robust SVD-based method[61].
3. Check the zero sets of the denominator functions. If they contain points in the region of interest near the origin, adjust the orders $N$ and $M$. We found that diagonal and near diagonal $[N \pm 2/N \pm 2]$ approximants tend to work the best.
4. Compute the coefficients of the numerators of the parametrization and of $\omega$ and $\kappa$ by evaluating (39) and (34).

Although Padé approximation is a well-researched topic, convergence results are generally limited. For meromorphic functions $f(z)$ with a finite number of poles at $z_1, \ldots, z_k$, the theorem of de Montessus de Ballore[66] guarantees the convergence of the approximants

$$[M/k](z) \text{ as } M \to \infty \tag{41}$$

globally. A stronger result is available for Stieltjes functions, i.e., for functions of the form

$$f(z) = \int_0^\infty \frac{d\mu(t)}{1+zt}, \tag{42}$$

for some positive measure $\mu$. In this case, the Padé approximants converge to $f(z)$ for all $z$ outside the negative real axis[24]. In the Supplementary Information, we discuss the Stieltjes-type center manifold of Euler's system[9,67–69], which, although non-analytic, can be described globally using Padé approximants.

## Rational function regression

Let us denote the function to be approximated as $\mathbf{f} : \mathbb{R}^d \to \mathbb{R}^\ell$. This could represent $\mathbf{f} = \mathbf{W}$ with $\ell = n$ or $\mathbf{f} = \mathbf{R}$ with $\ell = d$. We assume that the value of $\mathbf{f}$, denoted $\boldsymbol{\zeta}_i = \mathbf{f}(\boldsymbol{\eta}_i)$, is known at points $\boldsymbol{\eta}_i, \ldots, \boldsymbol{\eta}_K$ in the domain of interest. We then approximate $\mathbf{f}$ as

$$\mathbf{f}(\boldsymbol{\eta}) \approx [N/M](\boldsymbol{\eta}) = \frac{\sum_{|\mathbf{k}|=0}^{N} \mathbf{a}_{\mathbf{k}} \boldsymbol{\eta}^{\mathbf{k}}}{\sum_{|\mathbf{k}|=0}^{M} b_{\mathbf{k}} \boldsymbol{\eta}^{\mathbf{k}}}, \tag{43}$$

where we have chosen a common denominator with coefficients $b_{\mathbf{k}}$ for all components of $\mathbf{f}$, similarly to vector Padé approximants[45,46]. To avoid introducing singularities for the approximants, we require that the denominator is non-zero at all points $\boldsymbol{\eta}_i$. This is equivalent to requiring that the denominator is strictly positive.

The coefficients are determined by minimizing the error

$$\mathcal{E}_r = \sum_{i=1}^{K} \left| \boldsymbol{\zeta}_i - \frac{\sum_{|\mathbf{k}|=0}^{N} \mathbf{a}_{\mathbf{k}} \boldsymbol{\eta}_i^{\mathbf{k}}}{\sum_{|\mathbf{k}|=0}^{M} b_{\mathbf{k}} \boldsymbol{\eta}_i^{\mathbf{k}}} \right|^2, \text{ such that} \tag{44}$$

$$\sum_{|\mathbf{k}|=0}^{M} b_{\mathbf{k}} \boldsymbol{\eta}_i^{\mathbf{k}} \ge \delta \quad \text{for } i = 1, \ldots, K, \tag{45}$$

for some small $\delta > 0$. The constrained minimization problem is solved by gradient-based optimization methods[70]. The initial guess for the coefficients is obtained by solving the linearized problem, minimizing

$$\sum_{i=1}^{K} \left| \left( \sum_{|\mathbf{k}|=0}^{M} b_{\mathbf{k}} \boldsymbol{\eta}_i^{\mathbf{k}} \right) \boldsymbol{\zeta}_i - \sum_{|\mathbf{k}|=0}^{N} \mathbf{a}_{\mathbf{k}} \boldsymbol{\eta}_i^{\mathbf{k}} \right|^2, \tag{46}$$

subject to the same positivity constraint. This is a linear least-squares problem, which can be solved efficiently using the method of ref. 42, based on robust Padé approximation[61].

## Data availability

The data discussed in the manuscript is available publicly in the `globalized-SSM` repository at https://github.com/haller-group/globalized-SSM.

## Code availability

The code for the numerical implementation of the methods is available publicly in the `globalized-SSM` repository at https://github.com/haller-group/globalized-SSM.

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

## Acknowledgements
We are grateful to Mohammad Farazmand for sharing his code to simulate the Kolmogorov flow.

## Author contributions
B.K. and G.H. designed the research, analyzed the examples, and edited the manuscript. B.K. carried out the formal analysis, developed the numerical algorithm, and wrote the manuscript.

## Funding

## Competing interests
The authors declare no competing interests.
