## [Transparent Peer Review file · Nature Communications]

Globalizing Manifold-Based Reduced Models for Equations and Data

Corresponding Author: Dr Balint Kaszas

Version 0:

Reviewer comments:

Reviewer #1

(Remarks to the Author)

Please refer to the PDF file for my comments and evaluation. As a summary:

The paper is written well, and includes a significant contribution. The structure and the description of the method (especially rational approximation) require considerable revision.

(Remarks on code availability)

The code can be used to reproduce the examples from the paper. It is not very reusable in its current state, because many methods are not documented and the examples are contained in individual jupyter notebooks. There is no README file to explain the code base, but it is structured fairly easily, so it is no problem to navigate and understand the general structure. With minor familiarity with the python module system, it is easy to run and reproduce the examples.

I would recommend to clean up the repository (especially removing auxiliary `__pycache__` folders etc.), to add a README file describing the structure, and to document all classes and methods with proper docstrings. It would also be very good if the jupyter notebooks would not contain so many auxiliary methods, but that those would be separated into the main codebase and just imported in the notebooks.

Reviewer #2

(Remarks to the Author)

(Remarks on code availability)

Reviewer #3

(Remarks to the Author)

(Remarks on code availability)

Reviewer #4

(Remarks to the Author)

Report on NCOMMS-24-82595 "Globalizing Manifold-Based Reduced Models for Equations and Data"

by B. Kaszas and G. Haller

This article is concerned with spectral submanifold modelling (SSM) for dynamical systems, a technique developed by the authors and introduced in a sequence of papers. This current article is an important and crucial step forward as it alleviates convergence limitations in the earlier versions and presents methodology that produces globally convergent reduced models in latent space. The key step is the recasting of a Taylor series representation into a Pade representation, thus eliminating the occurrence of finite radii of convergence arising from complex-plane singularities.

The paper is very well organized and written. It motivates the need for rational approximations and presents various cases, both model-based and data-driven, to convince the reader about advantages of a Pade formalism.

There is little to criticize, the results speak for themselves. However, in the derivation and the methodology section, a few minor additions may make the manuscript more understandable for the reader.

Minor comments:

[1] In equation (2): U should be defined.

[2] In equation (3): is this really a double sum, or just a single sum with a max-degree-constant (like in equation (8))?

[3] Equation (8) represents a vector divided by a vector. A sentence on how this quotient is meant would be appreciated. I assume elementwise.

[4] An explanatory sentence should be added why the authors keep the same denominator polynomial.

[5] I am sure the reader would appreciate some explanation why the phase-space trajectory is obtained by *backward* integration.

[6] A sentence of the role of L in Kolmogorov flow would be great (periodicity of the zonal forcing).

[7] Most Pade approximants work best with a diagonal setup, using identical degrees for the numerator and denominator polynomials, which enforces a finite limit at infinity. Is there a reason for this?

[8] How many principal components have been used in the inverted-flag problem?

[9] Typo: above equation (33): apporximant -> approximant

In summary, this is a remarkable paper that certainly should be published after the minor issues above have been incorporated.

(Remarks on code availability)

Version 1:

Reviewer comments:

Reviewer #1

(Remarks to the Author)

I really appreciate the additional edits in the manuscript and the time to update the code. I no longer have any objections to publication, and congratulate the authors for their excellent work.

(Remarks on code availability)

The code is now well documented, even with additional unit tests. The data is also included in the repository, and my feedback on README etc. was followed adequately. I no longer have any objections regarding code and data availability.

Reviewer #2

(Remarks to the Author)

(Remarks on code availability)

Reviewer #3

(Remarks to the Author)

(Remarks on code availability)

Reviewer #4

(Remarks to the Author)

The authors have addressed all concerns and questions in a satisfactory manner. I am happy to recommend publication of the manuscript.

(Remarks on code availability)

Globalizing Manifold-Based Reduced Models for Equations and Data

Response to Reviewer 1

B. Kaszás, G. Haller

We thank the Reviewer for their detailed comments on our manuscript and for taking the time to check the codebase and give us feedback on it. Below, we reply to the issues raised and detail the modifications we made to address them. The Reviewer's comments are displayed verbatim in **bold text**, while quotes from the revised manuscript are *italicized*. We also attach a version of the manuscript in which the changes are highlighted with blue text.

Remarks on Code availability

The code can be used to reproduce the examples from the paper. It is not very reusable in its current state, because many methods are not documented and the examples are contained in individual jupyter notebooks. There is no README file to explain the code base, but it is structured fairly easily, so it is no problem to navigate and understand the general structure. With minor familiarity with the python module system, it is easy to run and reproduce the examples.

I would recommend to clean up the repository (especially removing auxiliary pycache folders etc.), to add a README file describing the structure, and to document all classes and methods with proper docstrings. It would also be very good if the jupyter notebooks would not contain so many auxiliary methods, but that those would be separated into the main codebase and just imported in the notebooks.

We appreciate that the Reviewer carefully checked the code we provided. We agree that the codebase needed additional work in order to be practically usable. Therefore, we have added documentation for the classes, cleaned up the file structure and reorganized the examples for improved readability. We also added a short README file with basic installation and usage instructions. The updated code repository is available in the same folder as before, under the link <https://polybox.ethz.ch/index.php/s/IjbFA0AmBe0XkxP>.

Major issues

i).The structure of the paper should be adjusted. In particular, leaving most of the mathematical details to Section 4 while referencing equations such as Equation 20 multiple times in Section 2 is confusing. The motivation to not overwhelm the reader with details and get a more abstract understanding of the methods and results first is laudable, but I believe the manuscript could be improved by moving Section 4.1. up to Section 2.1, and incorporate the first part of Section 4.2. (for example, up to equation 32) in Section 2.2. Generally, outlining the structure of the paper better at the end of the introduction would also help.

We thank the Reviewer for pointing this out. Indeed, our goal was to leave out the mathematical details from the Results section for improved readability. We understand, however, that referencing key equations in the Results section that are only introduced later, was confusing. To alleviate some of this confusion, we now state the autonomous invariance equation in the Results section. The paragraphs now read as follows:

In the autonomous limit with $\varepsilon = 0$, the d -dimensional ($d \leq n$) primary SSM, $\mathcal{W}(E)$, can locally be represented as

the image of a parametrization $\mathbf{W} : U \subset \mathbb{R}^d \rightarrow \mathbb{R}^n$, over some open set $U \subset \mathbb{R}^d$ as

$$\mathcal{W}(E) = \{\mathbf{x} = \mathbf{W}(\mathbf{p}) \mid \mathbf{p} \in U\} \subset \mathbb{R}^n. \quad (1)$$

The reduced dynamics $\dot{\mathbf{p}} = \mathbf{R}(\mathbf{p})$, with $\mathbf{R} : U \rightarrow \mathbb{R}^d$ are conjugate to (1), i.e., $\mathcal{W}(E)$ satisfies the invariance equation

$$\mathbf{A}\mathbf{W}(\mathbf{p}) + \mathbf{f}(\mathbf{W}(\mathbf{p})) = D\mathbf{W}(\mathbf{p})\dot{\mathbf{p}}. \quad (2)$$

We refer to Section 4.1 for a discussion on SSMs of the nonautonomous system with $\varepsilon > 0$.

We solve Eq. (2) by representing the parametrization of $\mathcal{W}(E)$ and its reduced dynamics as a power series truncated to some order N , i.e.,

$$\begin{aligned} \mathbf{W}^N(\mathbf{p}) &= \sum_{|\mathbf{k}|=0}^N \mathbf{W}_{\mathbf{k}} \mathbf{p}^{\mathbf{k}}, \\ \mathbf{R}^N(\mathbf{p}) &= \sum_{|\mathbf{k}|=0}^N \mathbf{R}_{\mathbf{k}} \mathbf{p}^{\mathbf{k}}. \end{aligned} \quad (3)$$

We define the multi index $\mathbf{k} = (k_1, \dots, k_d)$ and $|\mathbf{k}| = k_1 + k_2 + \dots + k_d$, so that $\mathbf{p}^{\mathbf{k}} = p_1^{k_1} p_2^{k_2} \dots p_d^{k_d}$ refers to a scalar monomial of the components of \mathbf{p} with total order $|\mathbf{k}|$. The coefficients $\mathbf{W}_{\mathbf{k}}$ and $\mathbf{R}_{\mathbf{k}}$ are vectors in \mathbb{R}^n and \mathbb{R}^d , respectively, for all \mathbf{k} .

The coefficients $\mathbf{R}_{\mathbf{k}}$ depend on the style of parametrization used. In the graph style parametrization, the reduced coordinates are obtained as projections onto the spectral subspace E , while in the normal form style parametrization, non-resonant terms are set to zero. The difference between these two choices is explained in more detail by, e.g., (Jain, Haller, 2022; Haro, et al. 2016; Stoychev, Römer, 2023).

We have also adopted your suggestion to outline the structure of the manuscript in the introduction. The final paragraph in the Introduction now reads as:

In the following Section 2, we present our results on using Padé approximants to construct reduced models on global SSMs (gSSMs). We discuss four examples, including the Kolmogorov flow (Chandler, Kerswell 2013), a nonlinear von Kármán beam in periodic and chaotic regimes (Jain, Haller, 2022; Liu et al., 2024), and the data-driven model of an inverted flag experiment (Xu et al. 2024). The mathematical details of SSMs and Padé approximants are discussed in Section 4. The Supplementary Material contains further applications and examples.

ii) What happens for nonautonomous systems? Much of the current focus in both experiments and the outline of the methods is devoted to autonomous systems. Although mentioned in certain parts of the manuscript, such as paragraph 2 of Section 4.1, and Example 2, it is not clear what the strengths and limitations of the new method are for nonautonomous systems. In addition, it is unclear from the manuscript what effect high-order nonautonomous corrections will have on this specific method, even though the authors claim they will have an improvement in the last sentence of Section 2.5. An added paragraph in Section 3 on the exact strengths and limitations of the proposed method regarding nonautonomous systems, differentiating between what has been shown experimentally and hypotheses based on previous research, would greatly improve the manuscript.

This is a good point. We have added a discussion on higher-order nonautonomous corrections in the Methods section, which reads as

As shown by Breunung, Haller, 2018; Ponsioen et al., 2019 and Thurnher et al., 2024, to first order in ε and to order- \hat{N} in the reduced coordinates, the resulting expression is

$$\mathbf{R}_{\varepsilon}(\mathbf{p}, \Phi) = \mathbf{R}(\mathbf{p}) + \varepsilon \sum_{|\mathbf{k}|=0}^{\hat{N}} \mathbf{S}_{\mathbf{k}}(\Phi) \mathbf{p}^{\mathbf{k}} + O(\varepsilon^2), \quad (4)$$

where the coefficients $\mathbf{S}_{\mathbf{k}}(\Phi)$ can also be Fourier-expanded. Accounting for the phase-dependence of the nonautonomous coefficients in the reduced dynamics increases the accuracy of the reduced models for higher forcing amplitudes, at the cost of computing the coefficients $\mathbf{S}_{\mathbf{k}}(\Phi)$ for each forcing frequency of interest. These computations are already implemented in `SSMTool`.

However, the phase-dependence of the parametrization and the reduced dynamics are often only small effects. It is, therefore, common to keep only the leading-order phase-dependent term in the reduced dynamics, resulting in ...

A forced gSSM-model can be constructed using multivariate Padé approximants based on the coefficients $\mathbf{S}_{\mathbf{k}}(\Phi)$. This, however, requires computing the coefficients for each forcing frequency separately. In the main text, we prefer to use only the leading-order approximation, which allows us to write down a closed-form expression for the forced response (26). We have decided to include a comparison showing the effects of the higher-order terms on the chaotic von Karman beam. The following text has been added to the last paragraph of Section 2.6.

The reduced model is in close agreement with the full model, even though the forced dynamics was approximated by simply projecting the forcing term onto the tangent space of the SSM according to (23). Incorporating higher-order corrections of the forced dynamics, as in (22), further improves the model, leading to more accurate short-time predictions. We present these comparisons, along with additional properties of the chaotic gSSM model, in the Supplementary Material.

The Supplementary Material now contains the following discussion:

The reduced dynamics on the SSM reads as

$$\mathbf{R}_\varepsilon(\mathbf{p}, \Phi) = \mathbf{R}(\mathbf{p}) + \varepsilon \sum_{|\mathbf{k}|=0}^{\hat{N}} \mathbf{S}_{\mathbf{k}}(\Phi) \mathbf{p}^{\mathbf{k}} + O(\varepsilon^2), \quad (5)$$

where $\hat{N} \geq 0$ is the approximation order for the forcing. In the main text, we have only considered the leading-order contribution with $\hat{N} = 0$, but the accuracy of the model can be improved by including higher-order terms as well. The corresponding gSSM-model can be constructed by computing an appropriate Padé-approximant for the forcing term in (5).

We compare predictions of the leading-order approximation presented in the main text and a [6/6] Padé-approximant computed with $\hat{N} = 17$ in Fig. 1. Qualitatively, the same type of chaotic dynamics is observed for the gSSM models, even when higher-order corrections are taken into account. However, for short times, the prediction error decreases even further when the nonautonomous terms are included. We also note that the polynomial SSM-model did not improve, even when we included the phase-dependent terms in (5). Specifically, the model experiences the same finite-time blowup as the leading-order approximation.

iii) It would be easier for the reader if the authors gave some more information regarding multivariate rational approximation presented in subsections 4.2 and 4.3. For example, what does it mean $\mathbf{z}^{\mathbf{k}}$ if \mathbf{z} and \mathbf{k} are vectors? are coefficients $a_{\mathbf{k}}$, $b_{\mathbf{k}}$ matrices?

We agree with this point. We now elaborate on the multi-index notation where it is first introduced.

We define the multi index $\mathbf{k} = (k_1, \dots, k_d)$ and $|\mathbf{k}| = k_1 + k_2 + \dots + k_d$, so that $\mathbf{p}^{\mathbf{k}} = p_1^{k_1} p_2^{k_2} \dots p_d^{k_d}$ refers to a scalar monomial of the components of \mathbf{p} with total order $|\mathbf{k}|$. The coefficients $\mathbf{W}_{\mathbf{k}}$ and $\mathbf{R}_{\mathbf{k}}$ are vectors in \mathbb{R}^n and \mathbb{R}^d , respectively, for all \mathbf{k} .

Minor issues

iv) P.3, It would be useful to define a map \mathbf{R} , similar to how $\mathbf{W} : \mathbb{R}^d \rightarrow \mathbb{R}^n$ is defined. The authors should also specify to which spaces the matrices \mathbf{W}_k and \mathbf{R}_k belong.

Figure 1: (a): Time series of the reduced coordinate η_1 on a chaotic trajectory of the full system (black). Higher-order nonautonomous corrections are included in both the SSM-reduced forced model (red) and the gSSM-reduced model (violet). The leading-order approximation of the gSSM-reduced model is also shown orange. (b): Relative error of the reduced-model for short times.

We agree that this information should be specified. We have added a definition for the map \mathbf{R} and the following sentences after eq. (3) to define the coefficients, as well as the multi index notation. We refer to our answer to question iii) above for the changes to the text.

v) P.3, formula (7), why is degree only even, $2n$, in the Taylor expansions?

Using the normal form style parameterization, the reduced dynamics only contain near-resonant terms. Specifically, for the case of two-dimensional SSM tangent to a spectral subspace with complex eigenvalues, the equation for \dot{p} only contains terms of the form $p^{k_1}\bar{p}^{k_1+1}$. When expressed in polar coordinates, this results in $\omega(\rho)$ and $\kappa(\rho)$ being even functions. We have added the following sentences before this equation to emphasize this point:

With the normal form style parametrization, the reduced dynamics only contain near-resonant terms of the form $p^{k_1+1}\bar{p}^{k_1}$ and $p^{k_1}\bar{p}^{k_1+1}$, and it is conveniently expressed in polar coordinates (Jain, Haller 2022)...

3. P.4, left column, 6th line from below, I think $l \geq 2$ is already enough.

Thank you. We have adjusted the formula.

4. There seems to be a lack of analysis of the first experiment (Section 2.4). The gSSM model seems to be sub-par when approximating the orbit from the red fixed point to the blue point. A comment on why this happens and further investigation by creating plots similar to Fig 1.b-c, but from red to blue fixed point, would be helpful.

The Reviewer is right to point this out. The heteroclinic orbit from ω_0 to ω_2 folds over the eigenspace E , and hence we cannot represent the entire manifold as a graph over E . We have modified Figure 1 to highlight this observation and added the following comment to the text.

We also observe that the manifold $\mathcal{W}(E)$ can only be represented as a graph over E for this segment, since the derivative $\frac{\partial}{\partial \xi} \mathbf{W}(\xi)$ diverges at a fold point near ω_0 . The parametrization, therefore, cannot be continued to capture ω_2 . Note, however, that the Taylor approximation diverges well before encountering this unremovable singularity of the graph-style parametrization.

5. Stating how the three dimensions were chosen when creating a 3D slice for visualization in Section 2.4. would be nice. Whether it is arbitrary or chosen explicitly.

We have chosen three Fourier modes having significant amplitudes along the heteroclinic orbits. We have adjusted the caption of Figure 1 to reflect this.

6. Adding a sentence on the assumption of analytic governing functions in the abstract shows the scope of the method and would be helpful for the reader.

Thank you for this suggestion. We agree with it, and since we also discuss a non-analytic example in the Supplementary Material, we have modified the first sentence of the abstract as follows

One of the very few mathematically rigorous nonlinear model reduction methods is the restriction of a dynamical system to a low-dimensional, sufficiently smooth, attracting invariant manifold.

7. Showing how the reduced SSM model compares to gSSM for the data-driven example, Example 4, could further the authors' claims. However, as the equation-driven examples highlight this very strongly already, it is not strictly necessary.

Thank you for this comment. A high-order polynomial approximation is also able to capture the reduced-dynamics of the inverted flag, therefore we believe that a direct comparison of predictions of SSM and gSSM-reduced models may not be very revealing. However, we have added the following sentences to highlight the advantages of data-driven gSSM reduction.

... A diagonal approximant also ensures that the reduced dynamics remains well-behaved outside the domain of the training data. As opposed to classical polynomial regression, the values of the approximants remain bounded or only grow mildly for large $|\boldsymbol{\eta}|$.

The data-driven rational approximation, on the other hand, requires only 60 coefficients, which is a significant reduction, for the cost of a slightly increased computational burden. In addition, the rational approximants extrapolate to larger domains in a more controlled way.

8. The claim in the first paragraph of page 12, stating that data-driven suffer less from singularities, is perhaps not completely justified. In combination with the statement: “preventing the rational functions from becoming singular is essential” raises the question of how much less they suffer without adding additional tools to the proposed methods. An explanation of why this is the case would be helpful for the reader.

We agree with this point as well. We have decided to include a short comparison between the polynomial, the constrained rational approximant and the unconstrained rational approximants in the Supplementary Material. Therefore, we have changed the relevant sentence to

As we show in the Supplementary Material, preventing the rational functions from becoming singular is essential for an accurate approximation.

The Supplementary Material now contains the following discussion:

In the main text, we have shown that rational function regression is effective in modeling the reduced dynamics on a low-dimensional SSM. Denoting the reduced coordinates by $\boldsymbol{\eta} \in \mathbb{R}^2$, we approximate the reduced dynamics as

$$\dot{\boldsymbol{\eta}}(\boldsymbol{\eta}) \approx [N/M](\boldsymbol{\eta}) = \frac{\sum_{|\mathbf{k}|=0}^N \mathbf{a}_{\mathbf{k}} \boldsymbol{\eta}^{\mathbf{k}}}{\sum_{|\mathbf{k}|=0}^M b_{\mathbf{k}} \boldsymbol{\eta}^{\mathbf{k}}}. \quad (6)$$

In addition, we require that the denominator is non-zero at all points $\boldsymbol{\eta}_i$ in the training set.

We then determine the coefficients by minimizing the error

$$\mathcal{E}_r = \sum_{i=1}^K \left| \zeta_i - \frac{\sum_{|\mathbf{k}|=0}^N \mathbf{a}_{\mathbf{k}} \eta_i^{\mathbf{k}}}{\sum_{|\mathbf{k}|=0}^M b_{\mathbf{k}} \eta_i^{\mathbf{k}}} \right|^2, \quad (7)$$

such that

$$\sum_{|\mathbf{k}|=0}^M b_{\mathbf{k}} \eta_i^{\mathbf{k}} \geq \delta \quad \text{for } i = 1, \dots, K, \quad (8)$$

for some small $\delta > 0$. We point out that without the regularization constraint (8), the regression can yield spurious singularities, which render the reduced model unusable in practice. To illustrate this, in Fig. 2 we compare the vector fields obtained by polynomial regression (SSM) and rational function regression (gSSM). Due to singularities in the domain of interest, unconstrained rational function regression is unable to recover the correct phase portrait. The polynomial SSM-model and the constrained gSSM-models both capture the dynamical features of the reduced vector field accurately. However, outside the range of the training data bounded by the stable limit cycle, the SSM-model starts to develop large gradients.

Figure 2: Comparison of data-driven models of the inverted flag experiment. (a): Reduced vector field on the SSM approximated by an order-11 polynomial. (b): The vector field is approximated by a [5/5] rational function without the constraint (8). (c): Same as (b), but the constraint (8) is enforced.

9. P.13, What does it mean 'Pade approximant of order N-M'? They usually write 'Pade approximant of type (N,M)'.

Thank you for pointing this out. We have adopted your suggestion.

10. There is something wrong with the formula (31). I think

$$\sum_{m=0}^n c_{n-m} b_m = 0, \quad n = N + 1, \dots, N + M,$$

for $N \leq M$ and

$$\sum_{m=0}^M c_{n-m} b_m = 0, \quad n = N + 1, \dots, N + M,$$

for $N \geq M$. Similar remarks regarding the formulas (30), (32) and (36) (what is n in (36)?) The authors have to check and correct them.

Thank you for catching these errors in the formulas. Indeed, we have checked and corrected these formulas. In particular (30), (31), (32) have been fixed as follows

This results in the linear equations for the coefficients a_n, b_n ,

$$a_n = \sum_{m=0}^n c_{n-m} b_m, \quad n = 0, 1, \dots, N + M. \quad (9)$$

We follow the robust approach of Gonnet et al. 2013, who first solve the homogeneous equations

$$\sum_{m=0}^n c_{n-m} b_m = 0, \quad n = N + 1, \dots, N + M, \quad (10)$$

with the convention of $b_j = 0$ for $j < 0$ or $j > M$. Using the singular value decomposition (SVD) of the Toeplitz matrix with elements $[c_{n-m}]$, the coefficients b_n can be computed. The remaining unknown coefficients a_n are then given as

$$a_n = \sum_{m=0}^n c_{n-m} b_m, \quad n = 0, 1, \dots, N. \quad (11)$$

11. P.14, left column, 15th line from above, 'The conditions (36) become'.

We have fixed this inconsistency.

12. P.14, § 4.3, for $\mathbf{f} = \mathbf{W}$ maybe $l = n$ instead of $l = p$ according to definition of $\mathbf{W}: \mathbb{R}^d \rightarrow \mathbb{R}^n$ on P.3. What is the relation between n and d ? There should be also some description / sentence before formula (41).

We are grateful to the Reviewer for pointing out these inconsistencies. We have fixed $l = n$ and added the assumption $d \leq n$ on Page 3. We have also adjusted the sentence before equation (41) to read:

We assume that the value of \mathbf{f} , denoted $\zeta_i = \mathbf{f}(\boldsymbol{\eta}_i)$, is known at points $\boldsymbol{\eta}_i, \dots, \boldsymbol{\eta}_K$ in the domain of interest. We then approximate \mathbf{f} as ...

13. It would be better to use ℓ instead of l and ℓ instead of l .

This is a good point. We have adopted your suggestion and have changed the notation of the indices.

14. P.14, formula (36), $|\mathbf{l}| = \mathbf{0}$ instead of $|\mathbf{l}| = 0$.

Thank you for noticing this. We have fixed the formula.

Globalizing Manifold-Based Reduced Models for Equations and Data

Response to Reviewer 4

B. Kaszás, G. Haller

We thank the Reviewer for their detailed reading of our manuscript. We are delighted that they found the paper well-organized and well-written. Below, we reply to the issues raised and detail the modifications we made to address them. The Reviewer's comments are displayed verbatim in **bold text**, while quotes from the revised manuscript are *italicized*. We also attach a version of the manuscript in which the changes are highlighted with blue text.

Minor issues

1) In equation (2): \mathbf{U} should be defined.

Thank you for pointing this out. We have added its description to the paragraph before Equation (2). It now reads as follows:

In the autonomous limit with $\varepsilon = 0$, the d -dimensional ($d \leq n$) primary SSM, $\mathcal{W}(E)$, can locally be represented as the image of a parametrization $\mathbf{W} : U \subset \mathbb{R}^d \rightarrow \mathbb{R}^n$, over some open set $U \subset \mathbb{R}^d$ as...

2) In equation (3): is this really a double sum, or just a single sum with a max-degree-constant (like in equation (8))?

Thank you for pointing out the inconsistencies in our notation. We have adjusted Eq. (3) to be consistent with the rest of the summation formulas in the paper.

3) Equation (8) represents a vector divided by a vector. A sentence on how this quotient is meant would be appreciated. I assume elementwise.

Equation (8) describes a ratio of two scalars, since the monomial terms $\mathbf{z}^{\mathbf{k}}$ are also scalars. We realize we have forgot to explicitly define this notation, so we have added the following sentence to the paragraph where it first appears, before Eq. (3).

We define $\mathbf{k} = (k_1, \dots, k_d)$ and $|\mathbf{k}| = k_1 + k_2 + \dots + k_d$, so that $\mathbf{p}^{\mathbf{k}} = p_1^{k_1} p_2^{k_2} \dots p_d^{k_d}$ refers to a scalar monomial of the components of \mathbf{p} with total order $|\mathbf{k}|$.

We have also added a reference to the notation after Eq. (8).

4) An explanatory sentence should be added why the authors keep the same denominator polynomial.

Thank you for pointing this out. We enforce the common denominator for the data-driven rational function regression, because it helps us avoid spurious singularities. In this way, only one denominator needs to be checked beforehand instead of all coordinate components. To explain this, we have added the following sentence to the end of Section 2.3.

In addition, we enforce that all components of the vector function share the same denominator and that the denominator is never zero on the training data. Having a common denominator for all components of the vector function makes it simpler to avoid spurious singularities.

5) I am sure the reader would appreciate some explanation why the phase-space trajectory is obtained by *backward* integration.

We have added the following explanatory sentence.

To verify the validity of the reduced-order models, predictions should be compared to trajectories of the full system. However, since the fixed point is stable, a nearby initial condition will leave its neighborhood along the heteroclinic orbit only in backward time. Therefore, we integrate the initial condition $\xi(0) = 10^{-5}$, close to the stable fixed point, backward under the SSM-reduced and the gSSM-reduced dynamics.

6) A sentence of the role of L in Kolmogorov flow would be great (periodicity of the zonal forcing).

The forcing wave number L primarily influences the bifurcation diagram and the dynamical characteristics of the turbulent flow for large Re . We have chosen $L = 4$ to be consistent with most related studies of the Kolmogorov flow. We have added the following sentence:

... where L denotes the forcing wave number. This influences the size of the large-scale flow structures, the bifurcations observed in (11) and the properties of the turbulent dynamics at high Re (Platt et al. 1991).

7) Most Padé approximants work best with a diagonal setup, using identical degrees for the numerator and denominator polynomials, which enforces a finite limit at infinity. Is there a reason for this?

Indeed, most applications use the diagonal Padé approximants. To our knowledge, there is no rigorous result on the optimality of the diagonal approximants for a given total order $N + M$. That being said, in the univariate case, diagonal Padé approximants correspond to continued fraction convergents, and the convergence-in-measure results also require the (generalization of the) diagonal setup. We have added the following sentence highlighting the connection with continued fractions:

Moreover, diagonal Padé approximants of a univariate function are related to the continued-fraction representation of the function.

However, the Reviewer's observation is also accurate: in a data-driven setting, $M \approx N$ ensures that the approximation is well-behaved outside the training data. Thank you for pointing it out. We have added the following sentence to the discussion of Example 2.7.

A diagonal approximant also ensures that the reduced dynamics remains well-behaved outside the domain of the training data. As opposed to classical polynomial regression, the values of the approximants remain bounded or only grow mildly for large $|\eta|$.

8) How many principal components have been used in the inverted-flag problem?

We have modified the relevant sentence to be more specific:

...we approximate the tangent space of the SSM at the fixed point using the two leading principal components of the delay-embedded trajectories.

9) Typo: above equation (33): apporximant \rightarrow approximant

Thank you for noticing this typo. We have fixed it.

Review report on the manuscript NCOMMS-24-82595 "Globalizing Manifold-Based Reduced Models for Equations and Data"

1 Summary

Restricting a dynamical system to a low-dimensional setting simplifies the nonlinear system, while capturing important properties of the system. This simplification can be crucial when modeling and often yields accurate predictive models. One type of such model is manifold-based reduction. In particular, under certain assumptions, attracting fixed points have convergent Taylor series in the neighborhood of the fixed point. However, current methods quickly blow up outside of a limited unknown convergence area, which is detrimental to its predictive power and the ability to enquire information about other fixed points captured by the SSM.

In the paper, the authors present the so-called “global theory of spectral submanifolds” (SSMs). To extend the range of applicability of SSM-reduced models, they replace Taylor series expansion of functions by their approximation via rational functions constructed using the Padé method. The authors show the advantages of the gSSM with respect to the SSM with several examples. Given analytic governing functions, the authors apply analytic continuation to the Taylor series, extending the domain where the model is well-behaved. This idea is justified by complex analysis. In particular, the paper applies Padé approximation in the equation-driven case and extends the regression step of the data-driven method to rational function regression. In several computational experiments, the authors show remarkable improvements, capturing much more of the underlying SSM. The paper is well-written, outlines a broad range of techniques and relevant literature, and is a valuable contribution.

Overall evaluation: The paper is written well, and includes a significant contribution. The structure and the description of the method (especially rational approximation) require considerable revision, detailed below.

2 Major issues

1. The structure of the paper should be adjusted. In particular, leaving most of the mathematical details to Section 4 while referencing equations such as Equation 20 multiple times in Section 2 is confusing. The motivation to not overwhelm the reader with details and get a more abstract understanding of the methods and results first is laudable, but I believe the manuscript could be improved by moving Section 4.1. up to Section 2.1, and incorporate the first part of Section 4.2. (for example, up to equation 32) in Section 2.2. Generally, outlining the structure of the paper better at the end of the introduction would also help.
2. What happens for nonautonomous systems? Much of the current focus in both experiments and the outline of the methods is devoted to autonomous systems. Although mentioned in certain parts of the manuscript, such as paragraph 2 of Section 4.1, and Example 2, it is not clear what the strengths and limitations of the new method are for nonautonomous systems. In addition, it is unclear from the manuscript what effect high-order nonautonomous corrections will have on this specific method, even though the authors claim they will have an improvement in the last sentence of Section 2.5. An added paragraph in Section 3 on the exact strengths and limitations of the proposed method regarding nonautonomous systems, differentiating between what has

been shown experimentally and hypotheses based on previous research, would greatly improve the manuscript.

3. It would be easier for the reader if the authors gave some more information regarding multivariate rational approximation presented in subsections 4.2 and 4.3. For example, what does it mean z^k if z and k are vectors? are coefficients a_k, b_k matrices?

3 Minor issues

1. P.3, It would be useful to define a map \mathbf{R} , similar to how $\mathbf{W} : \mathbb{R}^d \rightarrow \mathbb{R}^n$ is defined. The authors should also specify to which spaces the matrices \mathbf{W}_k and \mathbf{R}_k belong.
2. P.3, formula (7), why is degree only even, $2n$, in the Taylor expansions?
3. P.4, left column, 6th line from below, I think $l \geq 2$ is already enough.
4. There seems to be a lack of analysis of the first experiment (Section 2.4). The gSSM model seems to be sub-par when approximating the orbit from the red fixed point to the blue point. A comment on why this happens and further investigation by creating plots similar to Fig 1.b-c, but from red to blue fixed point, would be helpful.
5. Stating how the three dimensions were chosen when creating a 3D slice for visualization in Section 2.4. would be nice. Whether it is arbitrary or chosen explicitly.
6. Adding a sentence on the assumption of analytic governing functions in the abstract shows the scope of the method and would be helpful for the reader.
7. Showing how the reduced SSM model compares to gSSM for the data-driven example, Example 4, could further the authors' claims. However, as the equation-driven examples highlight this very strongly already, it is not strictly necessary.
8. The claim in the first paragraph of page 12, stating that data-driven suffer less from singularities, is perhaps not completely justified. In combination with the statement: "preventing the rational functions from becoming singular is essential" raises the question of how much less they suffer without adding additional tools to the proposed methods. An explanation of why this is the case would be helpful for the reader.
9. P.13, What does it mean 'Padé approximant of order $N - M$ '? They usually write 'Padé approximant of type (N, M) '.
10. There is something wrong with the formula (31). I think

$$\sum_{m=0}^n c_{n-m}b_m = 0, \quad n = N + 1, \dots, N + M,$$
 for $N \leq M$ and

$$\sum_{m=0}^M c_{n-m}b_m = 0, \quad n = N + 1, \dots, N + M,$$
 for $N \geq M$. Similar remarks regarding the formulas (30), (32) and (36) (what is n in (36)?). The authors have to check and correct them.
11. P.14, left column, 15th line from above, 'The conditions (36) become'.
12. P.14, § 4.3, for $\mathbf{f} = \mathbf{W}$ maybe $l = n$ instead of $l = p$ according to definition of $\mathbf{W} : \mathbb{R}^d \rightarrow \mathbb{R}^n$ on P.3. What is the relation between n and d ? There should be also some description / sentence before formula (41).
13. It would be better to use ℓ instead of l and $\boldsymbol{\ell}$ instead of \mathbf{l} .
14. P.14, formula (36), $|\mathbf{l}| = 0$ instead of $|\mathbf{l} = 0|$.